# Monitoring Behaviors of Broiler Chickens at Different Ages with Deep Learning

**DOI:** 10.3390/ani12233390

**Published:** 2022-12-02

**Authors:** Yangyang Guo, Samuel E. Aggrey, Peng Wang, Adelumola Oladeinde, Lilong Chai

**Affiliations:** 1School of Internet, Anhui University, Hefei 230039, China; 2Department of Poultry Science, University of Georgia, Athens, GA 30602, USA; 3College of Biosystems Engineering and Food Science, Zhejiang University, Hangzhou 310058, China; 4U.S. National Poultry Research Center, USDA Agricultural Research Service, Athens, GA 30605, USA

**Keywords:** poultry production, animal welfare, deep learning, behavior recognition, DenseNet

## Abstract

**Simple Summary:**

Animal behavior in the poultry house could be used as an indicator of health and welfare status. In this study, a convolutional neural network models (CNN) network model was developed to monitor chicken behaviors (i.e., feeding, drinking, standing, and resting). Videos of broilers at different ages were used to build datasets for training the new model, which was compared to several other deep learning frameworks in behavior monitoring. In addition, an attention mechanism module was introduced into the new model to further analyze the influence of attention mechanism on the performance of the network model. This study provides a basis for innovating approach for poultry behavior detection in commercial houses.

**Abstract:**

Animal behavior monitoring allows the gathering of animal health information and living habits and is an important technical means in precision animal farming. To quickly and accurately identify the behavior of broilers at different days, we adopted different deep learning behavior recognition models. Firstly, the top-view images of broilers at 2, 9, 16 and 23 days were obtained. In each stage, 300 images of each of the four broilers behaviors (i.e., feeding, drinking, standing, and resting) were segmented, totaling 4800 images. After image augmentation processing, 10,200 images were generated for each day including 8000 training sets, 2000 validation sets, and 200 testing sets. Finally, the performance of different convolutional neural network models (CNN) in broiler behavior recognition at different days was analyzed. The results show that the overall performance of the DenseNet-264 network was the best, with the accuracy rates of 88.5%, 97%, 94.5%, and 90% when birds were 2, 9, 16 and 23 days old, respectively. In addition, the efficient channel attention was introduced into the DenseNet-264 network (ECA-DenseNet-264), and the results (accuracy rates: 85%, 95%, 92%, 89.5%) confirmed that the DenseNet-264 network was still the best overall. The research results demonstrate that it is feasible to apply deep learning technology to monitor the behavior of broilers at different days.

## 1. Introduction

The world’s population is expected to reach 9.5 billion by 2050 and the requirement for animal products (e.g., meat, eggs, and milk) will be increased by 70% as compared to 2005 levels [1]. As indicated, it is challenging to improve animal production efficiency and product quality under limited natural resources (e.g., fresh water, feed and land), thus precision livestock/poultry production is critical for addressing the issue [2,3,4]. A key task of precision poultry production is monitoring animal behaviors for the evaluation of welfare and health status.

Animal behaviors can reflect their physical health, and is an important basis for precise animal farming management. For animal behavior monitoring, there are two primary methods: contact and non-contact. In the contact monitoring method, one or more sensors are attached to animals or planted into animals to complete the data collection, and then animal behavior recognition is realized using data analysis and modeling [5,6,7,8]. However, the contact method generates stresses in some animals and affect animal health and welfare due to the installation of sensors. Non-contact monitoring methods are usually based on animal images/videos or audio to extract relevant features and build behavior recognition models. In addition, computer vision technology has been widely used in non-contact monitoring in animal farming [9,10,11]. Traditional computer vision technology is used to extract artificially designed features (e.g., color, shape, and texture) from images or videos and combine machine learning algorithms for recognizing animal behaviors. The recognition accuracy depends on the feature extraction method. In addition, external factors such as complex scenes, occlusion between animals, and light intensity impact the extraction efficiency of animal features, thereby affecting the accuracy of behavior recognition [10].

In recent years, the development of deep learning has not only broken through the difficult problem of visual feature representation and improved the cognitive level of images and videos, but also accelerated the technological progress of computer vision technology in animal farming [12,13,14,15]. In the development and application of computer vision, convolutional neural network (CNN) has become one of most mainstream methods for monitoring animal information and as a decision support tool in the precision animal farming [16,17]. In poultry farming, Zhuang and Zhang [12] used a CNN model to detect and predict sick birds. Pu et al. [18] proposed a method based on automatic CNN to identify crowding behaviors of chicken group with 99.17% accuracy. Zhang and Chen [19] improved the network structure of ResNet and proposed a ResNet-FPN disease chicken recognition model. The results show that the recognition rate of the model on the test set is as high as 93.7%. Li et al. [20] used a CNN network model to detect the drinking behavior of chickens, with an overall accuracy of 88.2%.

However, most of the current research focused on a specific age of chickens, and few studies considered birds at different age stages. Machine learning models that include birds at different stage of growth and feeding periods will improve the management and decision-making of poultry farming. The objectives of this study were to (1) construct a behavioral dataset of broilers at different days to develop a deep learning model for broiler detection; and (2) compare the performance of popular CNN network models (e.g., ResNet-152, ResNeXt-101, EfficientNet-B4, DenseNet-264, ECA-DenseNet-264) on the new dataset. Broiler behaviors tested by the model include feeding, drinking, standing, and resting.

## 2. Materials and Methods

### 2.1. Experimental Setup and Data Collection

The image collection was conducted in a research broiler house (20 birds per pen) on the Poultry Research Farm at the University of Georgia, Athens, USA. Unless otherwise stated, the experimental setup and data were the same as previously published [9]. High definition (HD) cameras (PRO-1080MSFB, Swann Communications, Santa Fe Springs, CA, USA) were mounted on the ceiling (2.5 m above floor) to capture video (15 frame/s, 1440 pixels × 1080 pixels) for broilers from day 1 to day 50. The images of d2, d9, d16 and d23 were selected. In each stage, 300 images of each of the four broilers behaviors (feeding, drinking, standing, and resting) were segmented, totaling 4800 images. Figure 1 shows the example of broiler behaviors sample segmentation on d16.

### 2.2. Image Datasets Augmentation

To obtain more sufficient behavioral features, original videos were image augmented based on multi-pose and multi-angle situations of broilers. Firstly, 50 images were randomly selected from each category in the original dataset (about 800 images) as the testing dataset. Then, contrast enhancement by 20% and decrease by 20%, brightness enhancement by 20% and decrease by 20%, rotate 90°, 180° and 270°, Gaussian blur, Gaussian noise, a total of 9 enhancement methods were adopted to the remaining images in the original data set. After image augmentation processing, each day had 10,200 images, of which 200 original images were used as the testing set for four behaviors. Then, 10,000 images were divided into training set and validation set at 4:1 ratio. The information of behavior dataset is shown in Table 1.

### 2.3. CNN Network Models

CNN is the most popular framework applied in computer vision tasks and natural language processing. CNN is a multi-layered network that can learn features of a target to perform an autonomous detection [21]. At present, there are many versions of the CNN model, such as ResNet [22], EfficientNet [23] and DenseNet [24].

(1) ResNet: the core of the ResNet model is to establish a short-skip connection between the front layer and the back layer, which helps the back-propagation of the gradient during the training process, so that a deeper CNN network can be trained to achieve higher accuracy [22]. In addition, ResNeXt is a variant of ResNet, and the overall network structure of the two models is similar. In the ResNeXt model, the single-channel convolution is transformed into a multi-branch multi-channel convolution [25].

(2) EfficientNet: Tan et al. [23] analyzed the influence of the depth and width of the convolutional network and the size of the input image on the performance of the convolutional network and proposed a method for mixing model scales. By setting certain parameter values to balance the depth and width of the convolutional network and input image size, the performance of the convolutional network was improved. The EfficientNet can not only achieve State-of-the-art, but also achieve superior results when extended to other datasets in terms of transfer learning [23].

(3) DenseNet: The DenseNet model is similar to ResNet, but it establishes a dense connection between all the previous layers and the latter layers, and achieves feature reuse through the connection of features on the channel [24]. These characteristics allows DenseNet to achieve better performance than ResNet with fewer parameters and computational costs. Table 2 shows the common architectures of DenseNet. In this study, DenseNet-264 was selected.

At present, the attention model has been widely used in various types of deep learning tasks such as natural language processing, image recognition and speech recognition. Its purpose is to obtain more detailed information about the target of interest, while suppressing other unusable information [26]. Therefore, the efficient channel attention was introduced into the DenseNet model (ECA-DenseNet) to explore its performance on broiler datasets of different days.

In the current study, several popular models (ResNet-152, ResNeXt-101, EfficientNet-B4, DenseNet-264, ECA-DenseNet-264) were selected for comparative analysis. The models parameters and FLOPs are shown in Table 3, while image input size was 224 × 224.

### 2.4. Parameters Setting

All CNN comparison models were trained and tested on a GPU (Graphics Processing Unit) server that uses Python language and builds models based on the Pytorch 1.7.1 (Meta AI, Menlo Park, CA, USA) deep learning framework. The detailed equipment configuration information of the test is shown in Table 4.

### 2.5. Performance Evaluation

The accuracy, precision, recall and F1 score were used to evaluate the performance of the newly developed CNN models. Accuracy is the ratio of the number of correct predictions to the total number of input samples, precision shows the ability of the model to accurately identify the target, recall reflects the ability of the model to detect the target and F1 score is a reconciling means of precision and recall. TP, TN, FP and FN are the numbers of true positive samples, true negative samples, false positive samples and false negative samples, respectively. The relevant evaluation indicators are as follows:(1)$$\mathrm{Accuracy}=\frac{\mathrm{TP}+\mathrm{TN}}{\mathrm{TP}+\mathrm{FP}+\mathrm{TN}+\mathrm{FN}}\;$$
(2)$$\mathrm{Precition}=\frac{\mathrm{TP}}{\mathrm{TP}+\mathrm{FP}}\;$$
(3)$$\mathrm{Recall}=\frac{\mathrm{TP}}{\mathrm{TP}+\mathrm{FN}}\;$$
(4)$$F_{1}-\mathrm{score}=2\times \frac{\mathrm{Precision}\;\times \;\mathrm{Recall}}{\mathrm{Precision}+\mathrm{Recall}}\;$$

## 3. Results

### 3.1. Detection Results of the CNN Model

The recognition accuracy of the CNN models for the broiler datasets are illustrated in Table 5. In the dataset of d2, the DenseNet-264 achieved an accuracy of 88.5%, a precision of 88.8%, a recall of 88.5% and a F1 score of 88.6%, which was better than that of other comparison methods. In the dataset of d9, the recognition results of DenseNet-264 and ResNeXt-101 was the same (97%, 97.1%, 97%, 97%) and was superior to ResNet-152, EfficientNet-B4 and ECA-DenseNet-264. In the data set of d16, DenseNet-264 and ResNet-152 achieved the best recognition result (94.5%, 94.8%, 94.5%, 94.6%). In the data set of d23, DenseNet-264 had the best recognition result (90%, 89.9%, 90%, 89.9%), followed by ECA-DenseNet-264 (89.5%, 89.6%, 89.5%, 89.5%). Our findings illustrate that the DenseNet-264 obtained the best results on the datasets of broilers at different days, and its overall performance was also the best.

### 3.2. Results of Broiler Behavior Recognition

As it can be concluded by Table 5, ResNet-152, ResNeXt-101 and ECA-DenseNet-264 varied in their performances, while Densenet-264 was consistently the best for each dataset. To further analyze the performance of these four models for broiler behavior recognition, the confusion matrix is presented (Figure 2).

As indicated by Figure 2, standing and resting behaviors were misclassified by the models in the behavior datasets of d2 because of small sizes of broilers. In addition, some d23 broilers were misclassified as well because the fixed imaging system failed to collect information from multiple angles. This could be resolved by using a mobile imaging system in the future. For d9 and d16 behavior datasets, the results of feeding, drinking, standing and resting behavior were correctly recognized using the four CNN models. In addition, the behavior recognition results of Densenet-264 on the behavior datasets of d2, d9, d16 and d23 was better than other comparison methods, especially for standing and resting behavior recognition. In general, Densenet-264 was better at recognizing broiler behaviors at different growth stages compared to other models used in this study.

### 3.3. Detection Results of Densenet-264 Model

Compared with other models in the current study, Densenet-264 had the best detection effect for different age group of broilers. Therefore, the detection results of Densenet-264 was further analyzed. Figure 3 displays some behavior classification examples obtained by the Densenet-264 model. GT means ground truth.

Figure 3a shows that the broilers drinking-, feeding-, resting- and standing- behavior were correctly classified. In addition, the accuracies of drinking and feeding behaviors were higher than resting and standing behaviors, because these behaviors are distinctly characterized by contact with feeders or drinkers.

## 4. Discussion

### 4.1. CNN Model Performance Analysis

From the results, we can conclude that the DenseNet-264 model outperformed ResNet-152, ResNeXt-101 and EfficientNet-B4 in classifying broiler behaviors at different days. Densenet-264 had higher performance in detecting broiler behaviors because it considers the characteristics of the dense connection between all the previous layers and the latter layers. Therefore, the behavior features of broilers are not weakened with propagation.

From Table 5, it can be observed that the accuracy of the comparison algorithm is about 85%, while the accuracy of DenseNet-264 is 88.5% on the d2 dataset. Combined with Figure 2a, it can be observed that DenseNet-264, as a dense network, reduces the loss of features during feature propagation and improves the recognition of resting behavior and standing behavior. Additionally, the detection accuracy of each model in the data sets on d9 and d16 was improved, because the broiler was larger and the behavioral characteristics were more obvious due to the growth period. However, on the data set of d23, the accuracy of all models has decreased compared with that of d9 and d16. At this time, the broilers have grown up and are large, and appear crowded in a limited activity area (pen). In addition, their feathers are full, which leads to the similarity of some behavioral characteristics, such as resting behavior and standing behavior misidentification, as shown in Figure 2d.

In addition, attention mechanism enables the deep learning model to focus on the most relevant areas of the input to extract more discriminant features that have been widely used in computer vision and image processing [27]. The efficient channel attention was introduced into the DenseNet-264 model (ECA-DenseNet-264) to explore its performance on broiler datasets of different days. The results showed that ECA-DenseNet-264 did not improve the performance of broiler behavior recognition. This finding could be attributed to two possible reasons: (1) The network connection structure of DenseNet-264; (2) The input sample image size of 224 × 224 and low pixels, which makes the attention mechanism unable to effectively assign weights.

### 4.2. Data Analysis of Broiler Behavior Images

The broiler images contained broilers of different days, resulting in inconsistent size of the segmented images with the change of days when the broiler region was segmented. When the image was inputted into the CNN network, it was uniformly adjusted to 224 × 224, resulting in a wider or narrower broiler image. This adjustment affected the feature learning and behavior recognition effect of CNN network. In addition, there were multiple postures and angles of broilers in the scene that could lead to the misclassification of behaviors (e.g., Figure 3). Broiler behaviors were misclassified when the segmented image had a few feeders or drinkers. For example, the drinking behavior on d9 in Figure 3b was misclassified as a standing behavior. Additionally, when broilers are closer to feeders or drinkers but not feeding or drinking, misclassification of behaviors could occur (Figure 3b, d23). In addition, the shooting angle, broiler posture, and interference by objects can lead to misclassification of behaviors. For example, on d2, d9, d16, and d23 samples/images, the head of broilers were raising, lowering, or pecking feathers (Figure 3b), which resulted in the misclassification of the behaviors.

The data obtained from a single angle lacks the information of depth and other angles, and there was interference information such as background in the clipped sample images. If the edge contour of the broiler is used for segmentation, the detection accuracy can be improved. How to realize the recognition of multi-posture and multi-behavior of broilers in commercial production environment needs verification studies in commercial houses in the future.

## 5. Conclusions

This study evaluated methods for recognizing broiler behaviors using the CNN model framework (DenseNet-264 network) at different broiler ages. Results show that the model had the accuracy rates of 88.5%, 97%, 94.5%, and 90% on d2, d9, d16 and d23, respectively, which is better than other existing CNN models such as ResNet-152, ResNeXt-101, EfficientNet-B4 and ECA-DenseNet-264. The DenseNet-264 worked well due to the dense connection between all the previous layers and the latter layers, and broiler behavior features were not weakened with propagation. The behavior recognition performance of Densenet-264 was also higher than other comparison methods as the birds size/age changed (i.e., d2, d9, d16, and d23). This was especially shown for standing and resting behavior recognition.

Deep learning can be used to detect animal target areas well in complex breeding flock environments. Additionally, on the basis of detecting and segmenting target areas, it can further obtain target information or conduct behavior recognition research. In the future, a set of real-time target detection and behavior recognition models can be constructed in combination with target detection algorithms.

## Figures and Tables

**Figure 1 animals-12-03390-f001:**
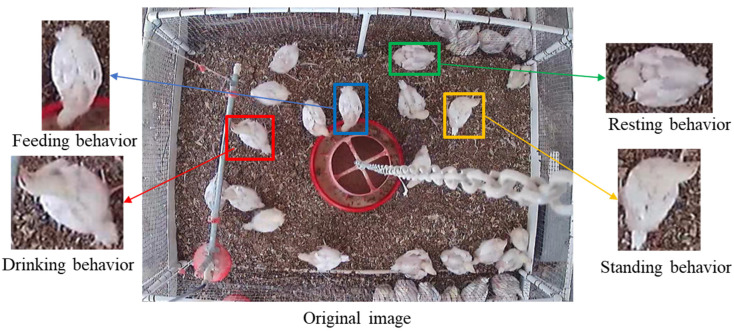
An example of broiler behaviors sample segmentation on d16.

**Figure 2 animals-12-03390-f002:**
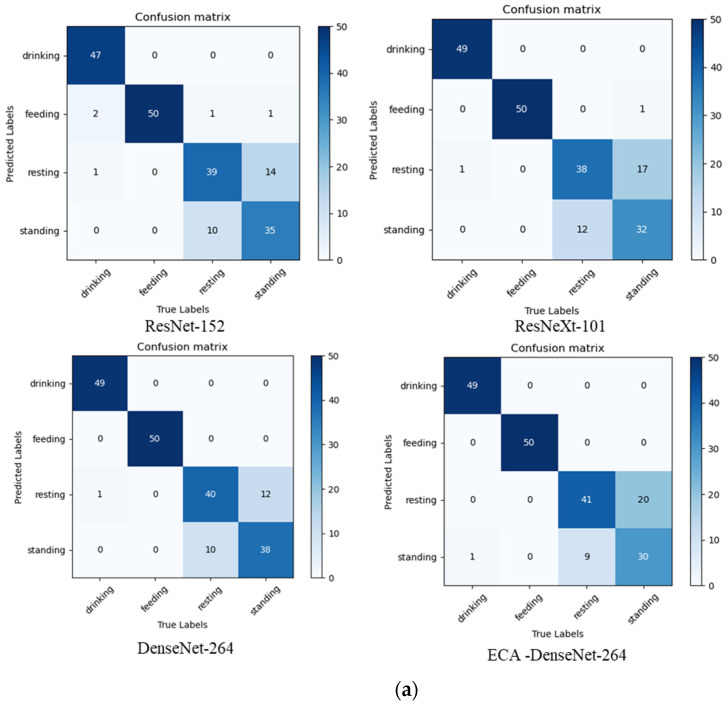
Confusion matrix of detection results on d2 (**a**), d9 (**b**), d16 (**c**), and d23 (**d**) datasets.

**Figure 3 animals-12-03390-f003:**
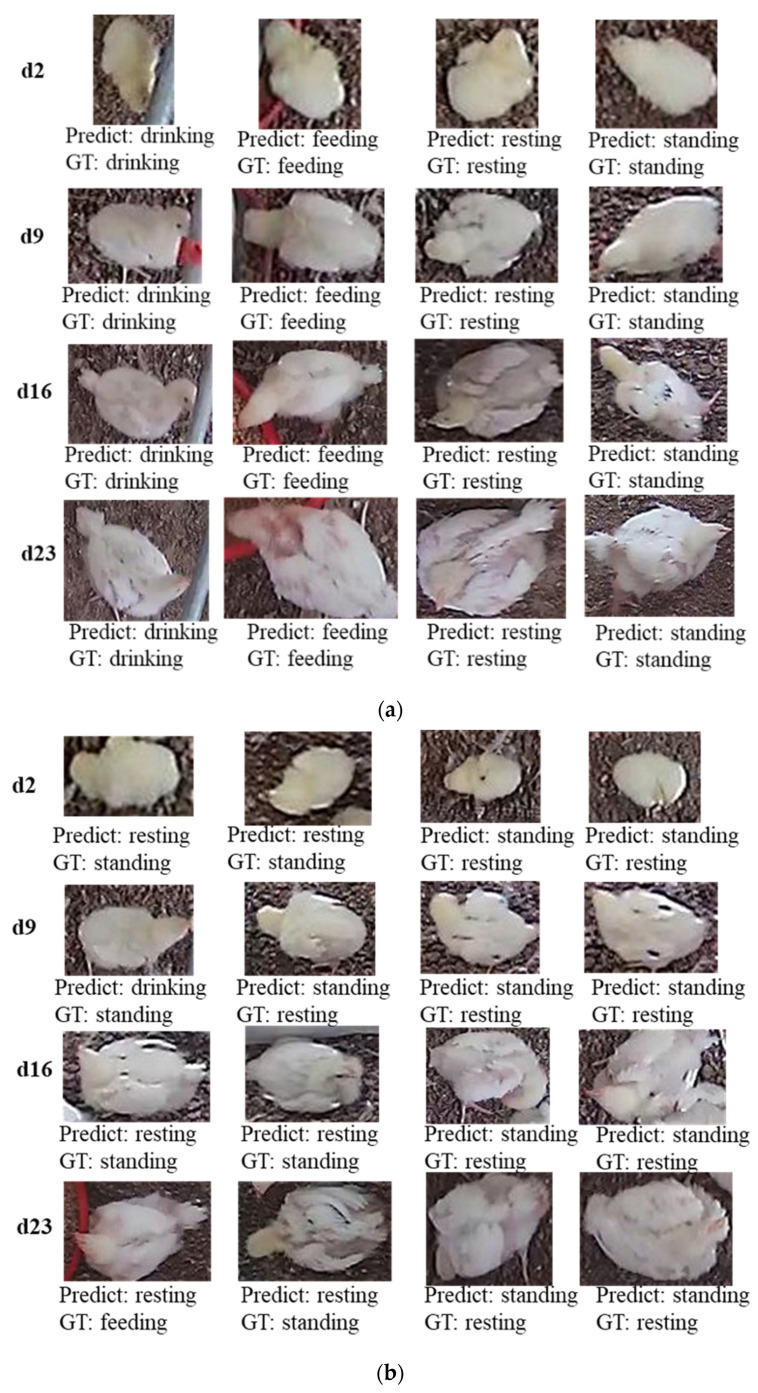
Behavior classification results of the Densenet-264 in broilers dataset. (**a**) True classification examples, (**b**) False classification examples.

**Table 1 animals-12-03390-t001:** Broiler behavior dataset information.

Categories	d2/d9/d16/d23 Dataset	Description
**Feeding**	2550/2550/2550/2550	Body is next to the feeder and the head is above the feed
**Drinking**	2550/2550/2550/2550	Head is close to and towards the drinker
**Standing**	2550/2550/2550/2550	Body is still, and the head may turn slightly
**Resting**	2550/2550/2550/2550	Body is close to the ground, and the head may turn slightly
**Training data**	8000/8000/8000/8000	--
**Validation data**	2000/2000/2000/2000	--
**Testing data**	200/200/200/200	--

**Table 2 animals-12-03390-t002:** Common architectures of DenseNet.

Layers	Output Size	DenseNet-121	DenseNet-169	DenseNet-201	DenseNet-264
Convolution	112 × 112	7 × 7 conv, stride 2			
Pooling	56 × 56	3 × 3 max pool, stride 2			
Dense Block (1)	56 × 56	$\left\{\begin{array}{c} 1\times 1\;\mathrm{conv}\\ 3\times 3\;\mathrm{conv} \end{array}\right\}\times 6$	$\left\{\begin{array}{c} 1\times 1\;\mathrm{conv}\\ 3\times 3\;\mathrm{conv} \end{array}\right\}\times 6$	$\left\{\begin{array}{c} 1\times 1\;\mathrm{conv}\\ 3\times 3\;\mathrm{conv} \end{array}\right\}\times 6$	$\left\{\begin{array}{c} 1\times 1\;\mathrm{conv}\\ 3\times 3\;\mathrm{conv} \end{array}\right\}\times 6$
Transition Layer (1)	56 × 56	1 × 1 conv			
	28 × 28	2 × 2 average pool, stride 2			
Dense Block (2)	28 × 28	$\left\{\begin{array}{c} 1\times 1\;\mathrm{conv}\\ 3\times 3\;\mathrm{conv} \end{array}\right\}\times 12$	$\left\{\begin{array}{c} 1\times 1\;\mathrm{conv}\\ 3\times 3\;\mathrm{conv} \end{array}\right\}\times 12$	$\left\{\begin{array}{c} 1\times 1\;\mathrm{conv}\\ 3\times 3\;\mathrm{conv} \end{array}\right\}\times 12$	$\left\{\begin{array}{c} 1\times 1\;\mathrm{conv}\\ 3\times 3\;\mathrm{conv} \end{array}\right\}\times 12$
Transition Layer (2)	28 × 28	1 × 1 conv			
	14 × 14	2 × 2 average pool, stride 2			
Dense Block (3)	14 × 14	$\left\{\begin{array}{c} 1\times 1\;\mathrm{conv}\\ 3\times 3\;\mathrm{conv} \end{array}\right\}\times 24$	$\left\{\begin{array}{c} 1\times 1\;\mathrm{conv}\\ 3\times 3\;\mathrm{conv} \end{array}\right\}\times 32$	$\left\{\begin{array}{c} 1\times 1\;\mathrm{conv}\\ 3\times 3\;\mathrm{conv} \end{array}\right\}\times 48$	$\left\{\begin{array}{c} 1\times 1\;\mathrm{conv}\\ 3\times 3\;\mathrm{conv} \end{array}\right\}\times 64$
Transition Layer (3)	14 × 14	1 × 1 conv			
	7 × 7	2 × 2 average pool, stride 2			
Dense Block (4)	7 × 7	$\left\{\begin{array}{c} 1\times 1\;\mathrm{conv}\\ 3\times 3\;\mathrm{conv} \end{array}\right\}\times 16$	$\left\{\begin{array}{c} 1\times 1\;\mathrm{conv}\\ 3\times 3\;\mathrm{conv} \end{array}\right\}\times 32$	$\left\{\begin{array}{c} 1\times 1\;\mathrm{conv}\\ 3\times 3\;\mathrm{conv} \end{array}\right\}\times 32$	$\left\{\begin{array}{c} 1\times 1\;\mathrm{conv}\\ 3\times 3\;\mathrm{conv} \end{array}\right\}\times 48$
Classification Layer	1 × 1	7 × 7 global average pool			
		1000D fully connected, softmax			

**Table 3 animals-12-03390-t003:** Params and FLOPs of CNN models.

	Methods	ResNet-152	ResNeXt-101	EfficientNet-B4	DenseNet-264	ECA -DenseNet-264
Parameters						
Params/M		60	84	19	34	68
FLOPs/B		11	32	4.2	6	12

**Table 4 animals-12-03390-t004:** Hardware and software systems.

Configuration Item	Value
**CPU**	Intel^®^ Xeon(R) Gold 5217 CPU@3.00 GHz
**GPU**	Nvidia Tesla V100 (32 GB)
**Operating System**	Ubuntu 18.04.5 LTS 64
**RAM**	251.4 GB
**Hard Disk**	8 TB

**Table 5 animals-12-03390-t005:** Detection results of CNN models for broiler dataset.

Days	Mothed	Accuracy	Precision	Recall	F1 Score
d2	ResNet-152	85.5	85.7	85.5	85.6
	ResNeXt-101	84.5	84.7	84.5	84.6
	EfficientNet-B4	85	85.2	85	85.1
	DenseNet-264	88.5	88.8	88.5	88.6
	ECA-DenseNet-264	85	85.6	85	85.3
d9	ResNet-152	94.5	94.6	94.5	94.5
	ResNeXt-101	97	97.1	97	97
	EfficientNet-B4	93.5	93.7	93.5	93.6
	DenseNet-264	97	97.1	97	97
	ECA-DenseNet-264	95	95.3	95	95.1
d16	ResNet-152	94.5	94.8	94.5	94.6
	ResNeXt-101	94	94.3	94	94.1
	EfficientNet-B4	90.5	90.9	90.5	90.7
	DenseNet-264	94.5	94.8	94.5	94.6
	ECA-DenseNet-264	92	92.5	92	92.2
d23	ResNet-152	89	89.2	89	89.1
	ResNeXt-101	89	89.2	89	89.1
	EfficientNet-B4	86.5	86.9	86.5	86.7
	DenseNet-264	90	89.9	90	89.9
	ECA-DenseNet-264	89.5	89.6	89.5	89.5

## Data Availability

Not applicable.

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
