# Peer review of "Monitoring Behaviors of Broiler Chickens at Different Ages with Deep Learning"

_animals, 2022, doi:10.3390/ani12233390_

Round 1

Reviewer 1 Report

Manuscript animals-2053739, entitled “Monitoring Behaviors of Broiler Chickens at Different Ages with Deep Learning”

Recommendation:       The above paper is not suitable for publication in its present form.

The article provides useful information about the performance of a network model on monitoring specific behaviors of broilers at different ages. Although, the experiment was in general appropriately designed and implemented, there are some points that should be corrected or clarified.

General comments

·       L81-85: This part is not appropriate for the introduction. Please remove

·       L208-216: I think that this part is more appropriate for Discussion. Please remove

·       Inadequate discussion. Please compare with other studies, as these in L67-73.

Minor points

L13: “…house could be used as an indicator of health and welfare status. In this study…”

L14: Please explain “CNN” here and not in L27.

L40: “As indicated” instead of “However”

L44: “…and health status.”

L69: What kind of behavior?

L75: “age stages” instead of “days”

L90: “were the same as” instead of “have been”

L105: “were” instead of “are”

Table 1 – Row of “Feeding”: “…is above the feed”

L126: Please delete “is”

L134: “selected” instead of “chosen”

Table 2: The rows “Dense Block (1)” and “Dense Block (2)” show similar data for all columns. Please check or merge

L142: “study” instead of “paper”

L144: “…in Table 3, while image input…”

L175: “Our findings” instead of “The results”

L177: “…was also the best.”

L180: “As it can be concluded by Table 5” instead of “From Table 4”

L183: “is presented” instead of “was obtained”

L189: “As indicated by Fig. 2…”

L190: Small sizes of what? Of broilers?

L192: “resolved” instead of “addressed”

L216: “…can also affect the…”

L230: “This finding could be attributed to two…”

L244: “ages” instead of “day”

L251: “shown” instead of “true”

Author Response

Reviewer 1:

The article provides useful information about the performance of a network model on monitoring specific behaviors of broilers at different ages. Although, the experiment was in general appropriately designed and implemented, there are some points that should be corrected or clarified.

Thanks for your comments.

General comments:

L81-85: This part is not appropriate for the introduction. Please remove

Response: Thanks for the suggestion. This part was removed as suggested.

L208-216: I think that this part is more appropriate for Discussion. Please remove

Response: Thanks, we have removed this part and expanded it in the Discussion section

Inadequate discussion. Please compare with other studies, as these in L67-73.

Response: Thanks for the suggestion. We compared the recognition results of each model on different datasets and the impact of data samples on the recognition results in the Discussion section. Lines 223-233 and Lines 250-262.

Minor points

L13: “…house could be used as an indicator of health and welfare status. In this study…”

Response: Thanks for the suggestion. we have revised it in the manuscript in Line 13.

L14: Please explain “CNN” here and not in L27.

Response: Thanks, we added the full name of CNN. Line 14.

L40: “As indicated” instead of “However”

Response: Thanks, we've replaced "however" with "As indicated"

L44: “…and health status.”

Response: Thanks for the suggestion. we have revised it in Line 45.

L69: What kind of behavior?

Response: The behavior is group crowding behavior, which has been added in Line 70.

L75: “age stages” instead of “days”

Response: Thanks, we've replaced "days" with "age stages". Line 76

L90: “were the same as” instead of “have been”

Response: Thanks, we've replaced "have been" with "were the same as". Line 88

L105: “were” instead of “are”

Response: Thanks, we've replaced "are" with "were ". Line 103

Table 1 – Row of “Feeding”: “…is above the feed”

Response: Thanks, we have revised it in Table 1.

L126: Please delete “is”

Response: Thanks, we have deleted it.

L134: “selected” instead of “chosen”

Table 2: The rows “Dense Block (1)” and “Dense Block (2)” show similar data for all columns. Please check or merge

Response: Thanks, we've replaced " chosen " with " selected ". Line 132

Each Dense Block group contains four groups of convolution layers for feature extraction to generate feature maps. We have redrawn Table 2.

L142: “study” instead of “paper”

Response: Thanks, we've replaced " paper" with "study". Line 140

L144: “…in Table 3, while image input…”

Response: Thanks, we have revised it in Line 142.

L175: “Our findings” instead of “The results”

Response: Thanks, we've replaced " The results " with " Our findings ". Line 173

L177: “…was also the best.”

Response: Thanks, we have revised it in Line 175.

L180: “As it can be concluded by Table 5” instead of “From Table 4”

Response: Thanks, we've replaced " From Table 4" with " As it can be concluded by Table 5". Line 178

L183: “is presented” instead of “was obtained”

Response: Thanks, we've replaced " was obtained " with " is presented ". Line 181

L189: “As indicated by Fig. 2…”

Response: Thanks, we have revised it in Line 187.

L190: Small sizes of what? Of broilers?

Response: Small sizes of broilers, we added it in Line 188

L192: “resolved” instead of “addressed”

Response: Thanks, we've replaced " addressed " with " resolved ". Line 190

L216: “…can also affect the…”

Response: Thanks, we have revised it.

L230: “This finding could be attributed to two…”

Response: Thanks, we have revised it in Line 231.

L244: “ages” instead of “day”

Response: Thanks, we've replaced " day " with " ages ". Line 257

L251: “shown” instead of “true”

Response: Thanks, we've replaced " true " with " shown ". Line 264

Reviewer 2 Report

Comments to the Authors of manuscript number: animals-2053739 entitled “Monitoring Behaviors of Broiler Chickens at Different Ages with Deep Learning”.

The study was performed on broiler at different ages whose behavior was analyzed using different methods.  The technique used is very much explained in many details.

 1. L 14- the abbreviation should be explained

2. L 81-85 – this part includes conclusion and results. It should be rephrased.

3. Figure 3 – “GT” should be explained

4. the discussion lacks other results obtained in this field. It could be deeper if Authors compare their results with others.

5. the deeper analysis of birds behavior is needed besides the technique used. The difference between all investigated days should be given. 

Author Response

Reviewer 2:

The study was performed on broiler at different ages whose behavior was analyzed using different methods.  The technique used is very much explained in many details.

Thank you for your comments.

  1. L 14- the abbreviation should be explained

Response: Thanks, we added the full name of CNN. Line 14.

  1. L 81-85 – this part includes conclusion and results. It should be rephrased.

Response: Thanks. We deleted this part.

  1. Figure 3 – “GT” should be explained

Response: Thanks for the suggestion. We have added “GT means ground truth” in Line 201.

  1. the discussion lacks other results obtained in this field. It could be deeper if Authors compare their results with others.

Response: Thanks for the suggestion. We compared the recognition results of each model on different datasets and the impact of data samples on the recognition results in the Discussion section. Lines 223-233 and Lines 250-262.

  1. the deeper analysis of birds behavior is needed besides the technique used. The difference between all investigated days should be given.

Response: Thanks for the suggestion. We discussed the differences between all investigated days and the impact on models detection results in the Discussion section. Lines 223-233

Round 2

Reviewer 1 Report

Authors made all the necessary amendments and I suggest the acceptance of their article.

Please use "observed" instead of "seen" in L435, 437

Reviewer 2 Report

I do not have any comments